# Transcriptomic and Metabolomic Analyses Reveal the Key Genes Related to Shade Tolerance in Soybean

**DOI:** 10.3390/ijms241814230

**Published:** 2023-09-18

**Authors:** Aohua Jiang, Jiaqi Liu, Weiran Gao, Ronghan Ma, Jijun Zhang, Xiaochun Zhang, Chengzhang Du, Zelin Yi, Xiaomei Fang, Jian Zhang

**Affiliations:** 1College of Agronomy and Biotechnology, Southwest University, Chongqing 400715, China; 2Institute of Specialty Crop, Chongqing Academy of Agricultural Sciences, Chongqing 402160, China

**Keywords:** soybean, shade tolerance, transcriptomic analysis, metabolomic analysis

## Abstract

Soybean (*Glycine max*) is an important crop, rich in proteins, vegetable oils and several other phytochemicals, which is often affected by light during growth. However, the specific regulatory mechanisms of leaf development under shade conditions have yet to be understood. In this study, the transcriptome and metabolome sequencing of leaves from the shade-tolerant soybean ‘Nanxiadou 25′ under natural light (ND1) and 50% shade rate (SHND1) were carried out, respectively. A total of 265 differentially expressed genes (DEGs) were identified, including 144 down-regulated and 121 up-regulated genes. Meanwhile, KEGG enrichment analysis of DEGs was performed and 22 DEGs were significantly enriched in the top five pathways, including histidine metabolism, riboflavin metabolism, vitamin B6 metabolism, glycerolipid metabolism and cutin, suberine and wax biosynthesis. Among all the enrichment pathways, the most DEGs were enriched in plant hormone signaling pathways with 19 DEGs being enriched. Transcription factors were screened out and 34 differentially expressed TFs (DETFs) were identified. Weighted gene co-expression network analysis (WGCNA) was performed and identified 10 core hub genes. Combined analysis of transcriptome and metabolome screened out 36 DEGs, and 12 potential candidate genes were screened out and validated by quantitative real-time polymerase chain reaction (qRT-PCR) assay, which may be related to the mechanism of shade tolerance in soybean, such as ATP phosphoribosyl transferase (ATP-PRT2), phosphocholine phosphatase (PEPC), AUXIN-RESPONSIVE PROTEIN (IAA17), PURPLE ACID PHOSPHATASE (PAP), etc. Our results provide new knowledge for the identification and function of candidate genes regulating soybean shade tolerance and provide valuable resources for the genetic dissection of soybean shade tolerance molecular breeding.

## 1. Introduction

Soybean (*Glycine max* L. Merr.) is an important legume crop worldwide, an important source of protein and oil and one of the most important cash crops [1,2]. The planting area of soybean has become larger and larger in the last 40 years. By 2050, soybean production will need to increase by 2.4% per year to meet market demand [3], which is a formidable challenge. Many researchers explored a range of methods to increase the soybean yield, such as intercropping of maize and soybean. However, in maize and soybean strip intercropping systems, due to the shading effect of the adjacent corn canopy, soybean plants receive less light and suffer from shade stress, which has a great negative effect on the growth and yield of soybean [4,5].

Light plays a vital role in the growth and development of plant. Photosynthesis is the material basis of crop yield formation and the most crucial process for obtaining organic products [6]. Crop yield is the product of photosynthesis and other interlinked physiological processes. The efficiency of light interception and material conversion is tightly bound to crop yields [7]. Shading influenced soybean growth and decreased starch deposition, particularly under heavy shading [8]. Owing to the influence of shading the growth of lateral branches of maize was inhibited and reduced vegetative biomass [9]. It was found that shading stress decreased photosynthetic production and grain yield in *Brassica rapa* [10]. Hence, it is of great significance to explore the response mechanism of shade stress and provide solutions for plant growth and development under light-limiting conditions.

In previous studies, it has been demonstrated that shade avoidance and tolerance responses in plants can be reflected in changes in the expression of genes, which were involved in hormone signaling and pigment biosynthesis [11]. It was also verified that the protochlorophyll reductase (*POR*), geranylgeranyl hydrogenase (*CHLP*), light-harvesting chlorophyll–protein complex II (*Lhcb1*), and ferredoxin (*N/A*) of porphyrin and chlorophyll metabolism were up-regulated expression under shade treatment of soybean seedling leaves [12]. It was confirmed by transcriptome sequencing that strong light inhibited the expression of the Photosystem II 10-kDa protein (PsbR) in *Baijiguan*, thereby affecting PSII stability, chloroplast development, and chlorophyll synthesis [13]. The photosensitive pigment interaction factor (PIF) regulated the expression of many genes, including metabolic enzymes, phytohormone signaling pathway, salicylic acid and jasmonic acid (JA), which suggested that it may be possible to improve crop growth at high planting densities by targeting the link between phyB and JA signaling [14]. At present, studies on shade avoidance response mainly focused on the model plant arabidopsis, rice, maize, wheat and other gramineous crops, but there are few studies on leguminous crops.

To explore the potential molecular mechanisms of shade tolerance in soybean, transcriptome and metabolome sequencing were performed on leaves of the shade-tolerant cultivar ‘Nanxiadou 25′ in 50% shade and without shading conditions. Combining functional gene analysis, DEGs analysis, weighted gene co-expression network analysis (WGCNA) and joint analyses of transcriptome and metabolome sequencing, we conducted a comprehensive investigation to explore the key regulatory factors and functional genes affecting the shade of soybean and understand the shade response mechanism, so as to provide valuable genetic resources for molecular breeding of soybean shade tolerance.

## 2. Results

### 2.1. Physiological Characteristics of Soybean Leaves under Different Shading Times

The physiological characteristics of soybean leaves related to photosynthesis were analyzed on different days without shading (ND) and shading (SHND) in Nanxiadou25. At 1 day (SHND1) and 5 days (SHND5) after shading, the activity of RuBisCO was higher than that of without shading (ND1 and ND5) (Figure 1), but at 9 days after shading (SHND9), it was significantly lower than that of without shading (ND9) (Figure 1). At 1, 5 and 9 days after shading treatment (SHND1, SHND5 and SHND9), the chlorophyll contents including total chlorophyll, chlorophyll a, chlorophyll b and carotenoid were significantly higher than that of the without-shading group (ND1, ND5, ND9) (Figure 1). Furthermore, the chlorophyll content increased gradually with the shading treatment time.

### 2.2. Transcriptome Sequencing and Analysis of DEGs in Soybean

According to the statistical analysis of physiological characteristics (Figure 1), high-throughput RNA-Seq of leaves 1 day after shading treatment (SHND1) and without shading (ND1) was performed (Figure 2A). The correlation evaluation of biological replicates showed a high correlation (Figure 2B). A total of 6 libraries were constructed and sequenced using the Illumina HiSeqTM 4000 platform. A total of 38.41 Gb of clean data was obtained from 6 samples. These clean reads were mapped to the reference genome with match ratios in the range of 94.23–96.19%. The percentage of Q30 bases in each sample ranged from 93.58% to 94.47%, and the GC content ranged from 44.21% to 44.66% (Appendix A). Through the differential expression analysis in ND1 vs. SHND1, a total of 265 differentially expressed genes (DEGs) were screened out, among which 144 DEGs showed down-expression level and 121 DEGs showed up-expression level after shading (Figure 2C, Appendix A). Hierarchical cluster analysis was performed for DEGs and DEGs with similar expression patterns may have the same function (Figure 2D).

### 2.3. KEGG Enrichment Analysis of DEGs

KEGG enrichment analyses showed that DEGs were significantly enriched in histidine metabolism (Ko00340), riboflavin metabolism (Ko00740), vitamin B6 metabolism (Ko00750), glycerolipid metabolism (Ko00561) and cutin, suberine and wax biosynthesis (Ko00073) (*q*-value < 0.05) (Figure 3A). A total of 22 differentially expressed genes were enriched in the top five pathways of KEGG (Figure 3B), including histidine metabolism, riboflavin metabolism, vitamin B6 metabolism, glycerolipid metabolism pathways, cutin, suberine and wax biosynthesis, most of which were down-regulated after shading.

Plant hormones regulate diverse processes in growth and development. In this study, there were most DEGs enriched in the phytohormone signaling pathway (Figure 3A). A total of 19 DEGs (14 up-regulated and 5 down-regulated) were involved in the phytohormone signaling pathway (Figure 4). Among these, 7 DEGs were involved in auxin metabolism, including 3 auxin-responsive proteins, 2 AUX/IAA family, 1 auxin transporter-like protein, 1 auxin response factor and 2 membrane transport protein, all of which were up-regulated expression after shading (Figure 4B). Six genes were involved in the cytokinin signaling pathway, with 5 up-regulated and 1 down-regulated gene. In addition, there were 1, 2, 2 and 1 differentially expressed genes included in the gibberellin, abscisic acid, jasmonic acid and salicylic acid metabolic pathways, respectively (Figure 4B).

### 2.4. Identification of Differentially Expressed Transcription Factors

A total of 7035 transcription factors (TFs) were screened, which were distributed in 20 families, mainly including *MYB* (330 genes), *AP2/ERF-ERF* (309 genes), *bHLH* (313 genes), *RLK-Pelle_DLSV* (278 genes), *C2H2* (271 genes), *WRKY* (186 genes) and *NAC* (180 genes) (Figure 5A). Furthermore, we identified 34 differentially expressed TFs (DETFs), with 21 up- and 13 down-regulated TFs in the SHND1 group (Figure 5B and Appendix A). Particularly, the two AUX/IAA genes (*Glyma.08G207900* and *Glyma.10G031900*) were identified, which were up-regulated after shading. Five genes, including 1 *MYB* (*Glyma.02G013900*), 2 *bZIP* (*Glyma.03G081700* and *Glyma.12G184400*), and 2 *bHLH* (*Glyma.13G368700* and *Glyma.17G058600*) were identified and were higher expression in ND group (Figure 5C). The potential DETFs may be involved in shade stress.

### 2.5. Identification of Genes Involved in Photosynthesis

To dig key genes involved in the shade stress of soybean, genes related to RuBisCO, photosystem I reaction center subunit (PSA) and chlorophyll a-b binding (CAB) genes were screened out and analyzed. A total of 30 genes related to RuBisCO and 29 genes related to PSA were identified and all these genes were not significantly different in ND1 vs. SHND1. A total of 47 CAB genes were identified and 2 genes (*Glyma.20G150600* and *Glyma.10G243800*) were down-regulated in the shading group (Figure 6C).

### 2.6. Weighted Gene Co-Expression Network Analysis (WGCNA)

All genes were analyzed using WGCNA to understand the regulatory network of shade stress in soybean, and three expression modules, MEbrown, MEblue and MEturquoise were obtained (Figure 7A). The correlation analyses of RuBisCO, chlorophyll a, chlorophyll b, carotenoids and total chlorophyll with three expression modules were performed (Figure 7B). Except for chlorophyll b, all traits were significantly negatively correlated with MEblue. Correlation analysis of genes with traits and modules showed that there were 20 core genes with MM (module membership) ≥ 0.95 and GS (gene significance) ≥ 0.95 in the MEblue module (Figure 7C,D). Cytoscape software was used to map the gene co-expression network (weight > 0.7) in the MEblue module and 26 hub genes were screened out (Figure 7E,F). Interestingly, all these core genes and hub genes had significant down-expression after shading. Among these, 10 DEGs simultaneously existed in Figure 7D,F, which could be key genes related to shade stress in soybean.

### 2.7. Metabolomic Analysis

Untargeted GC-MS analysis was performed in the leaves of ND1 and SHND1 and 101 differentially abundant metabolites (DAMs) were detected (Figure 8A, Appendix A), including 26 up-regulated metabolites and 75 down-regulated metabolites after shading. After log conversion of the difference multiple of differential metabolites in ND1 vs. SHND1, the top 10 metabolites, up-regulated and down-regulated in SHND1, are shown in Figure 8B. 3-Deoxyaconitine had the biggest increase (log2 fold charge = 15.6) and songorine had the biggest decrease (log2 fold charge = −18.91) in SHND1 (Figure 8B).

### 2.8. Combined Transcriptomic and Metabolomic Analyses

The KEGG enrichment analysis showed that DAMs and DEGs were strongly enriched in aspects of starch and sucrose metabolism, amino acid biosynthesis, carbon metabolism, galactose metabolism, and flavonoid biosynthesis. Among the top 10 KEGG-enriched pathways, the starch and sucrose metabolism pathway had the most DEGs, with 7 genes, and the ABC transporter pathway had 7 DAMs (Figure 9A). In all of the KEGG-enriched pathways, 36 DEGs were identified, including 20 up-regulated and 16 down-regulated genes (Figure 9B). To determine the correlation between DAMs and DEGs, correlation analysis was conducted based on the Pearson correlation coefficient. A total of 23 DEGs show a high correlation with DAMs (correlation coefficient > 0.8), and a network diagram showing the correlation between metabolites and genes was drawn (Figure 9C).

### 2.9. Screening and qRT-PCR Validation of Candidate Shattering Genes

Based on the above analysis, including KEGG enrichment, phytohormone signaling pathway, DETFs, WGCNA and combined analysis of transcriptome and metabolome sequencing, 12 candidate DEGs were screened out, which appeared in multiple analyses and had high expression levels (FPKM > 5). In order to verify the expression pattern of DEGs obtained by RNA-Seq analysis, qRT-PCR was used to detect the expression levels of 12 DEGs in ND1 vs. SHND1. Detailed primers for qRT-PCR analysis are listed in Appendix A. The expression levels of these DEGs by qRT-PCR were basically consistent with the FPKM values by RNA-Seq (Figure 10). These results confirmed that the transcriptome analysis data are reliable.

## 3. Discussion

Soybean is an important crop with abundant protein, vegetable oil and several phytochemicals. Cultivated soybean has a long history because of such predominant values. However, low-light environments are unfavorable for the growth of soybean. Numerous studies have confirmed that some soybean varieties have specific adaptive responses to shading conditions [11,15]. Shade-tolerant varieties have no or very thin stratum corneum, large leaf area, lush leaves, and few stomata and chloroplasts [16,17]. Previous research indicates that in the regular intercropping system of corn and soybean, soybean suffered a short light supply, reduced from 30% to 50%, compared to normal soybean planting [18]. In this study, the shade-tolerant soybean ‘Nanxiadou 25′ was selected as the experimental material and experimental group (SHND) under sunshade nets with a 50% shading rate and the control group (ND) without shading was set to explore the key genes involved in the shading response in soybean and reveal the molecular mechanism of soybean resistance to shading.

Chlorophyll content was directly related to the shade stress of plants [19]. It has previously been reported that carotenoids in green leaves ensure efficient photosynthesis, remove various reactive oxygen species, and protect chlorophyll from photooxidation [20]. In this study, the chlorophyll content including total chlorophyll, chlorophyll a, chlorophyll b and carotenoid was significantly higher than that of the without-shading group (ND1, ND5, ND9). Furthermore, the chlorophyll content increased gradually with the shading treatment time, which was consistent with previous studies.

Many plants have evolved adaptive mechanisms to cope with abiotic stress. Many transcription factors (TFS) play important roles in these adaptive mechanisms [21]. Light activates many transcription factors, such as MYB, bZIP, bHLH, WRKY, Zinc-finger, GATA, etc., which bond to light-responsive elements to modulate transcription [22]. Studies have found that the bHLH transcription factor also regulates plant response to environmental changes, such as controlling the light response and interacting with the circadian clock [23,24]. In this study, 34 TFs, including MYB, bHLH, WRKY, and NAC, were differentially expressed between both groups. Among them, one *MYB* (*Glyma.02G013900*), two *bZIP* (*Glyma.03G081700* and *Glyma.12G184400*), and two *bHLH* (*Glyma.13G368700* and *Glyma.17G058600*) were down-regulated in the SHND1 group, which could be involved in shade stress in soybean.

Hormones are signal molecules that regulate plant growth and development [25]. Most studies have shown that AUX, CTK and GA play important roles in plant growth [26]. Transcriptome analysis of young leaves under two light environments in *Cryptocarya concinna* showed that the genes related to hormones had up-regulated expression under low light conditions [27]. In our study, nine auxin-related genes and five gibberellin-related genes had significantly higher expression in the SHND1 group. This finding indicates that the low light environment significantly affected the expression of hormone-related genes in the soybean. It means that high expression of those genes may be beneficial to plant growth and response to low light environment.

WGCNA is an important tool for rapidly processing multiple transcriptome data and rapidly mining key genes highly related to traits [28,29]. It has been widely used in the study of morphogenesis and developmental regulation mechanisms of soybean [30,31,32,33]. In this study, all genes were analyzed by WGCNA and three expression modules were obtained, among which MEblue was highly correlated with physiological traits. In the MEblue module, 10 core hub genes were identified. Among them are three *phosphorylcholine phosphatases* (*PEPC*), which play a role in the absorption of CO_2_ in the atmosphere during the photosynthetic process of C4 and asparaginic acid metabolism [34,35]. In addition, there were two *MGD2*, *Glyma.13G162600* and *Glyma.17G108700*, which could be involved in galactolipids biosynthetic pathway and the photosynthetic membranes contain large amounts of galactoselipid MGDG [36,37]. It was speculated that *Glyma.13G162600* and *Glyma.17G108700* would be involved in the MGDG signaling pathway and photosynthesis of soybean.

In our research, the transcriptome and metabolome sequences were conducted to obtain novel insight into the molecular mechanisms for shading conditions. The results confirm that shade levels strongly affect phenotypic changes and physiological responses in soybean. The expression levels of TFs, auxin-related genes, gibberellin-related genes and many metabolites were significantly altered and exerted important functions in response to shade stress.

## 4. Materials and Methods

### 4.1. Plants and Sample Preparation

This study was conducted in 2022 at the experimental base of Southwest University in Chongqing, China. The shade-tolerant soybean ‘Nanxiadou 25′ obtained from the Nanchong Academy of Agricultural Sciences (Nanchong, China) was selected as the experimental material. The seeds were sown on 10 May, in 12 rows with a length of 2 m, row spacing of 0.5 m and plant spacing of 0.2 m. After 40 days of sowing, 6 rows were selected for shading treatment under sunshade nets with a shading rate of 50% (named SHND group) and the remaining 6 rows were selected for control (named ND group). At 1, 5 and 9 days after shading treatment, 10 soybean plants were randomly selected as biological samples from the control group and the treatment group, respectively, and a total of 3 biological replicates were set up for physiological indicators, transcriptomics and metabolomics analysis. The third upper leavers were taken from the main stems at 9:00 am that day. As in the previous study [38,39], we first measured physiological indicators including total chlorophyll, chlorophyll a and b content, carotenoid content and ribulose bisphosphate carboxylase oxygenase (RuBisCO). According to the results of the physiological indicators, the samples after 1 day of shading treatment were selected for transcriptomic and metabolomic analyses.

### 4.2. Physicochemical Properties

The RuBisCO activity was determined by plant ribulose 1,5-diphosphate carboxylase/oxygenase (RuBisCO) ELISA kit, according to the manufacturer’s procedure.

The chlorophyll content was determined using a spectrophotometer, and the specific operations are as follows. Soybean leaves with 0.1 g (m) were cut into pieces, fully ground in 10 mL ethyl alcohol (V) and placed in dark conditions until the leaves turned white completely. The light absorption values of the chlorophyll–ethanol solutions were determined at 470, 645 and 663 nm by spectrophotometry, and the total contents of chlorophyll a, chloroform b, chlorophyll and carotenoid concentration were calculated according to the following formula [40]:Chlorophyll a (mg/g) = (12.7 × A663 − 2.69 × A645) × V ÷ m ÷ 1000,
Chlorophyll b content (mg/g) = (22.9 × A645 − 4.68 × A663) × V ÷ m ÷ 1000, 
Total chlorophyll content (mg/g) = (20.21 × A645 + 8.02 × A663) × V ÷ m ÷ 1000, 
Carotenoid concentration (mg/g) = [(1000A470 − 3.27Ca − 104Cb) ÷ 229] × V ÷ m ÷ 1000.

### 4.3. RNA-Seq Analysis

The total RNA was extracted using a Trizol reagent (Invitrogen, Carlsbad, CA, USA) following the manufacturer’s procedure. Total RNA quantity and purity were measured with Bioanalyzer 2100 and RNA 6000 Nano LabChip Kit (Agilent, Santa Clara, CA, USA) with RIN number > 7, followed by gel extraction with 1% agarose gel electrophoresis. Then approximately 10 μg of total RNA was purified using poly-T oligo-attached magnetic beads and cleaved into smaller fragments with fragmentation buffer. Then the cleaved RNA fragments were transcribed to first-strand cDNA fragments using reverse transcriptase and a high concentration of random hexamer primer. The cDNA library was developed using the protocol for the RNA-Seq sample preparation kit (Illumina, San Diego, CA, USA). The average insert size for the paired-end libraries was 150 bp (50 bp). The mRNA library of each sample was constructed and sequenced in the Illumina HiSeq4000 platform. The adaptor and low-quality sequence were removed using Fastp with default parameters for raw data of sequencing [41], and clean reads were then mapped to the soybean genome using HISAT2 [42]. Differential expression analysis was performed between two different groups by DESeq2 and edgeR under the following standard parameters: false discovery rate (FDR) < 0.05 and |log2(ratio)| ≥ 1 [43]. When the Bonferroni (Q) corrected *p*-value was ≤0.05, the GO term and KEGG pathway analysis results were considered significant. The enriched KEGG pathway was determined using R 4.1.2, which was also used to construct a scatterplot of the results. Furthermore, some key DEGs associated with shade tolerance according to the previous research were used to construct a heatmap. The RNA-seq data discussed in this publication were deposited in NCBI (PRJNA951453).

### 4.4. Weighted Gene Co-Expression Network Analysis (WGCNA)

The WGCNA software package (version 1.6.6) in the R program was used to construct weighted gene co-expression networks for ND1 and SHND1. After entering the normalized gene expression matrix, SoftThreashold in the WGCNA package was used to calculate the weighted value. The blockwiseModules were used to construct scale-free networks, with default parameters. The soft connectivity function was used to calculate the connectivity degree of genes for obtaining the expression modules. The correlation analysis of three expression modules with shade-related traits (RaBisCO, chlorophyll a, chlorophyll b, carotenoid and total chlorophyll) was conducted. Cytoscape (version 3.9.1) was used to visualize networks in the module and screen out core genes [44]. The selection of core genes is based on the following factors. The correlation between a gene and other genes or its presence regulatory networks. If the gene is related to more genes exceeding the threshold, or node location genes are in the regulatory network, then genes are core genes and the threshold for the correlation coefficient is >0.95.

### 4.5. Metabolite Profiling Analysis

Samples (100 mg) were individually ground with liquid nitrogen, and the homogenate was re-suspended with pre-chilled 80% methanol, and 0.1% formic acid by vortexing. UHPLC-MS/MS analysis was conducted by using a Vanquish UHPLC system (Thermo Fisher, Dreieich, Germany) coupled with an Orbitrap QExactive^TM^ HF mass spectrometer (ThermoFisher). The raw data were obtained by using the Compound Discoverer 3.1 (CD3.1; Thermo Fisher) to conduct peak alignment, peak picking and quantitation analysis for each metabolite. The normalized data were used to predict the molecular formula based on the additive ions, molecular ion peaks and fragment ions. The metabolite quantitation applied an MRM pattern according to a previous study [45].

Meanwhile, the mixed samples were used as the quality control for monitoring the consistency of replicates from the extraction to the detection process. The detected metabolites’ peak areas were normalized by R (www.r-project.org/ accessed on 10 May 2023). Then, the normalized data were used to conduct a heat map analysis by R program. The differentially expressed metabolites analyses were conducted using the following standards: VIP > 1, a *p*-value < 0.05, and fold change ≥ 2 or <0.5. The significantly differential metabolites were subsequently submitted to PCA and KEGG enrichment analysis.

### 4.6. Validation of Differentially Expressed Genes by qRT-PCR Analysis

The qRT-PCRs were carried out using SYBR Premix Ex Taq II (Tli RNaseH Plus) in a volume of 20 μL of the reaction system, which contained 200 ng cDNA template, 0.5 mM of each of the forward and reverse primers, 10 µL BlastTaq™ 2× qPCR MasterMix, lastly nuclease-free water was added up to 20 μL. The PCR amplification procedure was as follows: 95 °C for 3 min; followed by 40 cycles at 95 °C for 15 s; and 58 °C for 30 s in a Bio-Rad CFX manager 2.0. Relative expression levels were estimated based on the PCR cycle threshold of the 2^−∆∆Ct^ method [46]. The values of 3 independent biological replicates and 3 technical replicates were averaged.

## 5. Conclusions

In this study, we generated transcriptome and metabolome sequencing of leaves from the shade-tolerant soybean ‘Nanxiadou 25′ under natural light (ND1) and 50% shade rate for one day (SHND1) and performed comprehensive transcriptomic analysis, including DEGs analysis, KEGG enrichment analysis, identification of DEGs involved in plant hormone signaling pathways, TF and photosynthesis, WGCNA and combined analysis of transcriptome and metabolome. Based on the above analysis and gene function, 12 candidate DEGs were finally screened out and validated by qRT-PCR, such as ATP phosphoribosyl transferase (ATP-PRT2), phosphocholine phosphatase (PEPC), AUXIN-RESPONSIVE PROTEIN (IAA17), PURPLE ACID PHOSPHATASE (PAP), etc., which deserve more attention for response mechanisms of soybean shading in the future. This study provides valuable resources for the molecular mechanism of soybean shade tolerance and the utilization of shade-tolerant soybean molecular breeding.

## Figures and Tables

**Figure 1 ijms-24-14230-f001:**
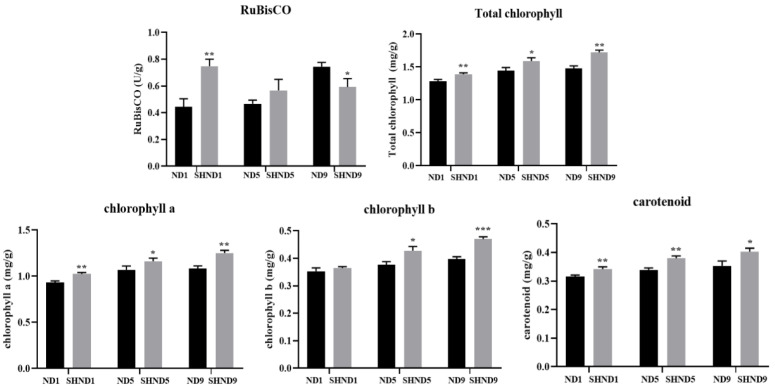
Bar chart of total chlorophyll, chlorophyll a and chlorophyll b, carotenoid content, and the activity of RuBisCO. Asterisks indicate the level of statistical significance (*, ** and *** represent signifificance at the 0.05, 0.01 and 0.001 probability levels).

**Figure 2 ijms-24-14230-f002:**
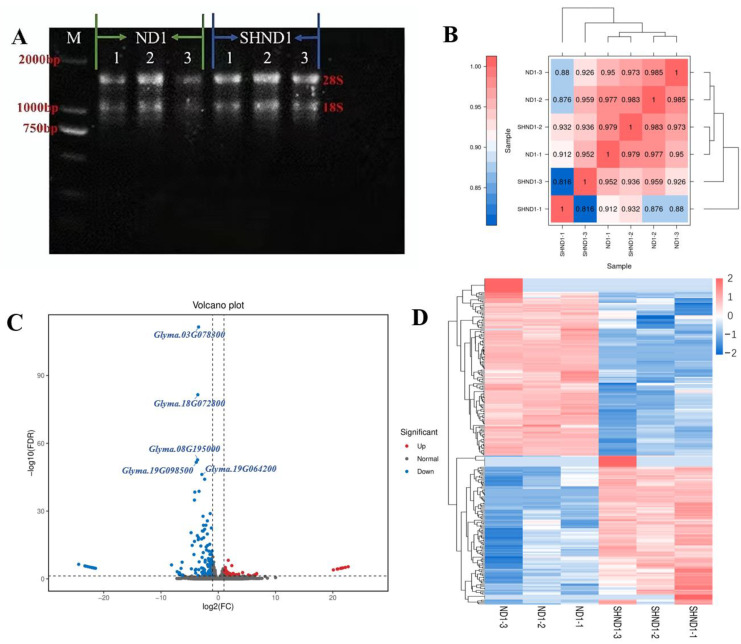
Transcriptome sequencing and analysis of DEGs in soybean. (**A**) Results of agarose electrophoresis of total RNA from samples. (**B**) Correlation analysis between samples. (**C**) The volcano map of DEGs. Each dot represents one gene, with blue dots representing down-regulated DEGs, red dots representing up-regulated DEGs, and gray dots representing non-differentially expressed genes. (**D**) Cluster map of DEGs. The color represents the level of gene expression in the sample log10 (FPKM+0.000001).

**Figure 3 ijms-24-14230-f003:**
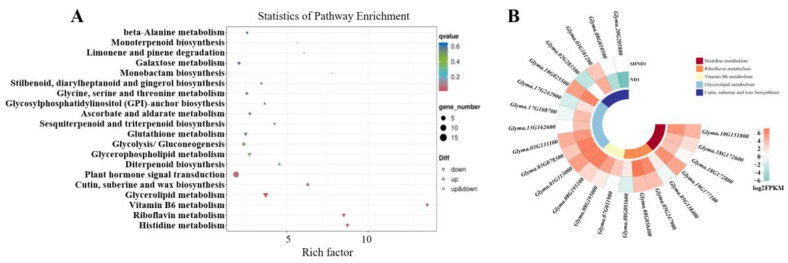
KEGG enrichment analysis of DEGs. (**A**) KEGG-enriched bubble map of DEGs. The higher the enrichment factor, the more significant the enrichment level of DEGs in this pathway. The color of the circle represents *q*-value. The size of the circle indicates the number of genes enriched in the pathway. (**B**) Circle heatmap of DEGs in top 5 KEGG enrichment pathways.

**Figure 4 ijms-24-14230-f004:**
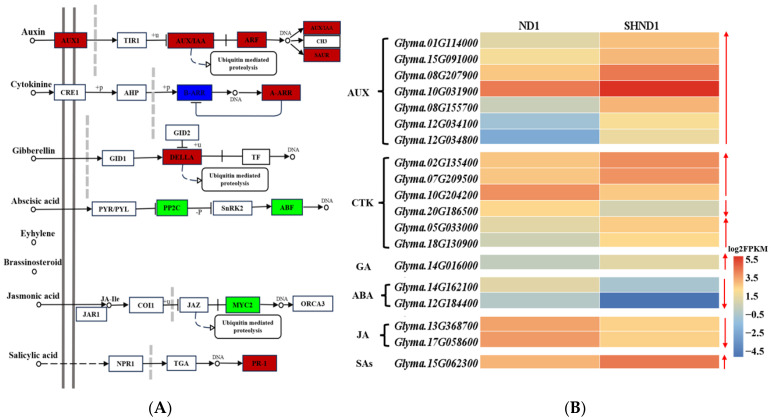
Analysis of DEGs involved in phytohormone signaling pathway. (**A**) Plant hormone signal transduction pathway. Red boxes mean up-regulated expression genes, green boxes mean down-regulated expression genes, and blue boxes mean up- and down-regulated genes. (**B**) Heatmap of DEGs involved in phytohormone signaling pathway.

**Figure 5 ijms-24-14230-f005:**
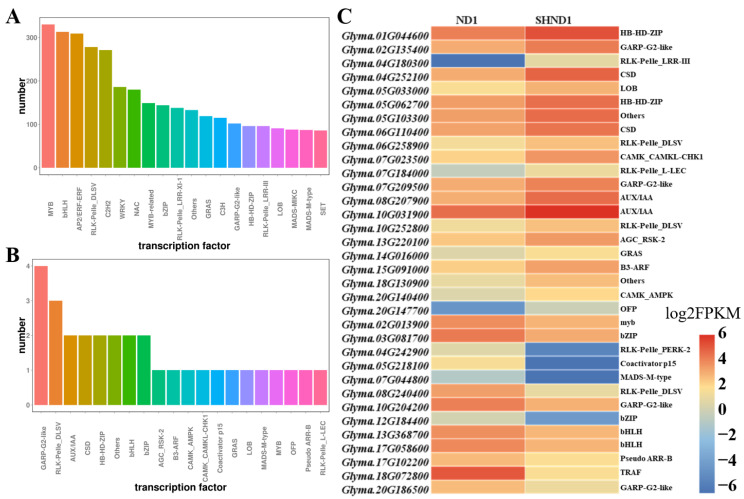
Transcription factor analysis. The number of identified TFs (**A**), DETFs (**B**), and heatmap of the differentially expressed transcription factors (**C**).

**Figure 6 ijms-24-14230-f006:**
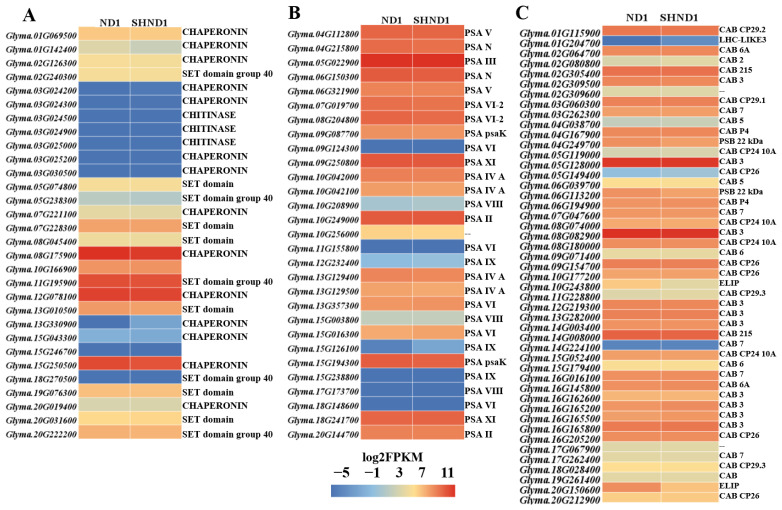
Heatmap of genes involved in photosynthesis. (**A**) RuBisCO-related genes, (**B**) photosystem I reaction center subunit (PSA), (**C**) chlorophyll a-b binding (CAB) genes.

**Figure 7 ijms-24-14230-f007:**
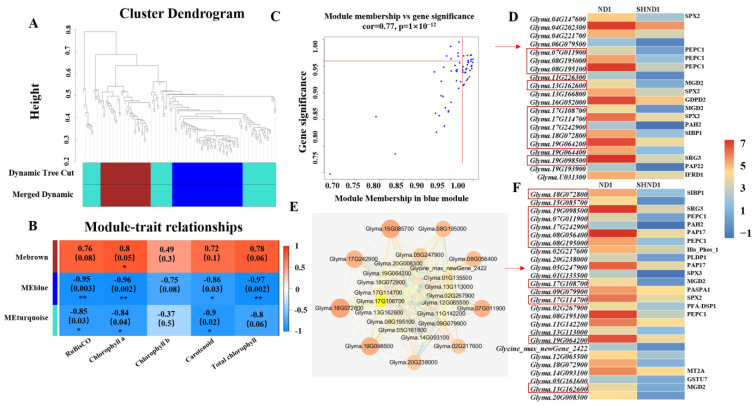
WGCNA co-expression network analysis. (**A**) Gene cluster dendrograms and module detecting. (**B**) Association analysis of gene co-expression network modules with traits. (* and ** represent signifificance at the 0.05 and 0.01 probability levels.) (**C**,**D**) Gene co-expression network and heatmap of core genes in MEblue module. The scattered points in the red box were core genes with MM (module membership) ≥ 0.95 and GS (gene significance) ≥ 0.95. (**E**,**F**) Gene co-expression network and heatmap of hub genes in MEblue module.

**Figure 8 ijms-24-14230-f008:**
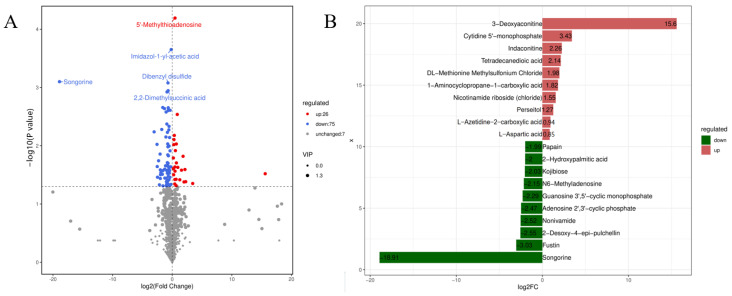
Analysis of metabolome sequencing of leaves in ND1 and SHND1. (**A**) Volcano plot of different metabolites. (**B**) The 20 most different metabolites in the comparison groups in colors red (up-regulated) and green (down-regulated).

**Figure 9 ijms-24-14230-f009:**
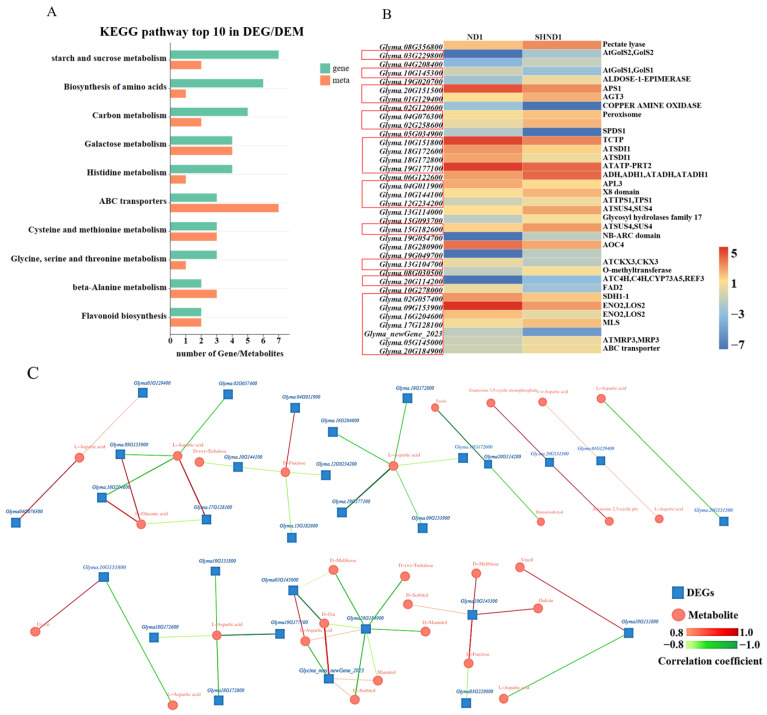
Combined analysis of transcriptome and metabolome. The top 10 pathways for KEGG enrichment analysis of common DAMs and DEGs (**A**). Heat map of the differential genes in the pathway, with red boxes indicating DEGs that are more than 0.8 correlated with DAMs (**B**). A network diagram of correlations between metabolites and genes (**C**).

**Figure 10 ijms-24-14230-f010:**
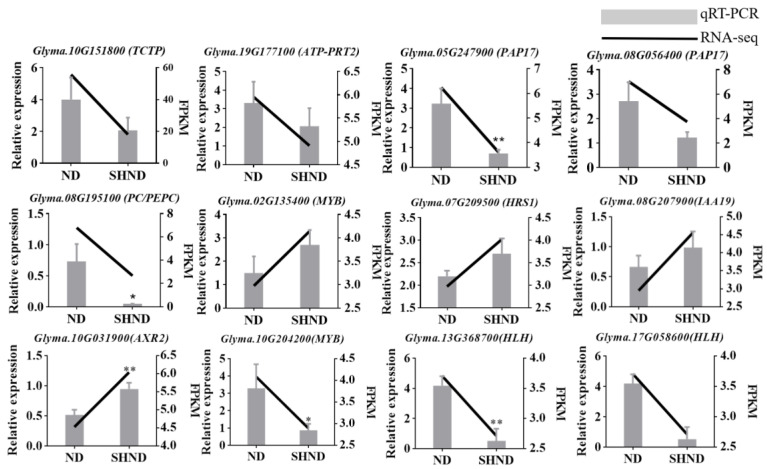
qRT-PCR validation of 12 soybean shade-tolerance-related differentially expressed genes. Bar graphs represent relative expression determined by qRT-PCR and line graphs represent expression levels determined by RNA-seq (FPKM). Error bars represent the standard error of three independent biological and three technical replicates of the qRT-PCR data. (* and ** represent signifificance at the 0.05 and 0.01 probability levels.)

## Data Availability

The datasets used and/or analyzed during the current study are available from the corresponding author upon reasonable request.

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
