# Peer review of "Transcriptomic and Metabolomic Analyses Reveal the Key Genes Related to Shade Tolerance in Soybean"

_ijms, 2023, doi:10.3390/ijms241814230_

Round 1

Reviewer 1 Report

In the submitted manuscript by Xiaomei Fang, Jian Zhang and colleagues entitled “Transcriptomic and metabolomic analyses reveal the key genes related to shade tolerance in soybean”, the authors studied the changes of transcriptome and metabolome under shade conditions just 1 day after treatment. To my opinion, you need to perform metabolomic analysis in additional time points (5 and 9 days after shading) to reinforce your outcomes. Overall, the manuscript is well-written, and the figures have a good presentation. However, there are some issues that should be carefully addressed.

Major and minor comments,

Please clarify in the MM section:

What time of day 'give a time period’ did you perform leaves sampling?

What about the size of the leaves, does it uniform? 

The authors mention ‘Three replicates were randomly selected’ How many leaves were included in each replicate from how many plants?

As the experiment was carried out in environmental conditions the author should also provide hourly temperature and relative humidity in order to perform the experiment from other groups that may be interested in the experimentation process.

Why you choose 50 % shade net? Do you conduct a pre-experiment to define the optimum shade net?

What about temperature under the net, certainly both temperature and RH will have been influenced. Do you have any data to support that environmental conditions (except shading) did not alter? In case you miss these data, you should discuss this probability.

You perform Weighted Gene Co-expression Network Analysis (WGCNA) in a single time point, you should support this option with a reference because WGCNA mainly used in time-points or developmental experiments.

Why do you choose to perform metabolomic analysis only 1 day after shading? You expected similar differences in the rest time points, you should support your decision.

Result section:

Line 208: The authors mention ‘DEMs’ before the abbreviation for the first time, you should mention the expression, in this case probably they mean 'differentially expressed metabolites (DEMs)' which is definitely wrong. The right expression is ‘differentially abundant metabolites (DAMs)’. So, replace DEM with DAM in the whole manuscript.

Author Response

Response to Reviewer 1 Comments

1.What time of day 'give a time period’ did you perform leaves sampling?

Reply:At 1, 5 and 9 days after shading treatment, the third upper leavers were taken at 9:00 am that day for the physiological indexes, transcriptomic and metabolomic analyses. This section has been added to the article.

2.What about the size of the leaves, does it uniform?

Reply:In this study, the third upper leavers from main stems of plants with uniform growth were selected as experimental samples. So, the size of third upper leavers from main stems of different plants was also uniform in this study.

3.The authors mention ‘Three replicates were randomly selected’ How many leaves were included in each replicate from how many plants?

Reply:The third upper leavers of the main stem of 10 soybean plants with uniform growth were randomly selected from the control group and the treatment group as a biological sample, and three biological replicates were set up.

4.As the experiment was carried out in environmental conditions the author should also provide hourly temperature and relative humidity in order to perform the experiment from other groups that may be interested in the experimentation process.

Reply:Sorry, the environmental data wasn’t collected in this study, considering that the treatment group and the control group were in the same climatic and soil conditions, which would not affect the results of this experiment.

5.Why you choose 50 % shade net? Do you conduct a pre-experiment to define the optimum shade net?

Reply:The purpose of this study is mainly to provide a theoretical basis for soybean corn strip compound planting. Previous research indicating that in the regular intercropping system of corn and soybean, soybean suffered short light supply reduced from 30% to 50%, comparing to the normal soybean planting. As a result, a shade net with a 50% shade rate was chosen for this study. Most of the current treatments for soybean shade research have used 50% shade netting. (Liu, B. C., Wang, Y., Li, S., Jin, L., Herbert, S. J. (2010). Soybean yield and yield component distribution across the main axis in response to light enrichment and shading under different densities. - Plant Soil Environment 56(8): 384-392.)

6.What about temperature under the net, certainly both temperature and RH will have been influenced. Do you have any data to support that environmental conditions (except shading) did not alter? In case you miss these data, you should discuss this probability.

Reply:The temperature and RH under the shading net were different from those of the control group. In order to be closer to the actual production, the temperature and RH of the treatment group and the control group were not completely controlled.

Liu, B. C., Wang, Y., Li, S., Jin, L., Herbert, S. J. (2010): Soybean yield and yield component distribution across the main axis in response to light enrichment and shading under different densities. – Plant Soil Environment 56(8): 384-392.

7.You perform Weighted Gene Co-expression Network Analysis (WGCNA) in a single time point, you should support this option with a reference because WGCNA mainly used in time-points or developmental experiments.

Reply:Relevant literature has been inserted

Karimi, M., Pakdel, MH., Lashaki, KB., and Soorni, A. Identification of hub salt-responsive genes in Cucumis sativus using a long non-coding RNA and mRNA interaction network. Horticulture, Environment, and Biotechnology (2022) 63:539–556

8.Why do you choose to perform metabolomic analysis only 1 day after shading? You expected similar differences in the rest time points, you should support your decision.

Reply:Based on physical and chemical properties between ND group and SHND group), we found that the physiological indexes at 1 day after shade were the most different from the control group, and the differences gradually decreased after 5 and 9 days. Therefore, sample after 1 day after shade were selected for transcriptome and metabolome sequencing.

Result section:

9.Line 208: The authors mention ‘DEMs’ before the abbreviation for the first time, you should mention the expression, in this case probably they mean 'differentially expressed metabolites (DEMs)' which is definitely wrong. The right expression is ‘differentially abundant metabolites (DAMs)’. So, replace DEM with DAM in the whole manuscript.

Reply:Replacement changes have been made in the article.

Reviewer 2 Report

Comments, 

1)    Line 13. I don’t think word sequence suits here.

2)    Line 19. What do you mean by statement “The phytohormone signaling pathways were analyzed, in which 19 DEGs were enriched”.

3)    Avoid short forms in abstract.

4)    Line 98-100. Author needs to re-phrase this sentence. It is meaningless. 

5)    Extensive editing of English language is necessary.

6)    Figure 2C. Please name at least 5 most significant genes in volcano plot.

7)    Figure 2D. What are all the genes in this clusters? If you are not naming, then this plot is meaningless.

8)    The paper is not well written. Although the paper has some interesting results, but the scientific (English) language used in paper is poor. Many sentences need re-phrasing. Take help of native English speaker.

9)    Line 348. How libraries prepared? Describe

10) What parameters of HISAT2? Star is better aligner than HISAT2. Why HISAT2 used here?

11) Section 4.3 re-write. Need to describe all the analyses.

12) Figure 9C. What is the significance of network?

13) Discussion section needs to be re-written. It’s not going with flow of story.

The paper is not well written. Although the paper has some interesting results, but the scientific (English) language used in paper is poor. Many sentences need re-phrasing. Take help of native English speaker.

Author Response

Response to Reviewer 2 Comments

  • Line 13. I don’t think word sequence suits here.

Reply:For the “shade-tolerant”, we referred the previous literature.

Bin C,Li W,Ranjin L, et al. Shade-Tolerant Soybean Reduces Yield Loss by Regulating Its Canopy Structure and Stem Characteristics in the Maize–Soybean Strip Intercropping System. Frontiers in Plant Science,2022,13.

  • Line 19. What do you mean by statement “The phytohormone signaling pathways were analyzed, in which 19 DEGs were enriched”.

Reply:Since shade tolerance is related to plant hormones, and the most DEGs were enriched in plant hormone signaling pathways with 19 DEGs in this study.

  • Avoid short forms in abstract.

Reply:The full name has been used when it first appeared.

  • Line 98-100. Author needs to re-phrase this sentence. It is meaningless. 

Reply:We had corrected as “According to the statistical analysis of physiological characteristics (Figure 1), high-throughput RNA-Seq of leaves at 1 day after shading treatment (SHND1) and control (ND1) was performed (Figure 2A)”.

  • Extensive editing of English language is necessary.

Reply:English grammar and other issues have been touched up.

  • Figure 2C. Please name at least 5 most significant genes in volcano plot.

Reply:The top five significant DEGs have been annotated in the volcano map

  • Figure 2D. What are all the genes in this clusters? If you are not naming, then this plot is meaningless.

Reply:Figure 2D shows the comparison of all DEGs expression levels between ND1 and SHND1, and the up-regulation and down-regulation of DEGs can be intuitively seen. There are too many DEGs to be clearly presented in the figure. The ID of genes and the naming of new genes are shown in supplementary Table 2.

  • The paper is not well written. Although the paper has some interesting results, but the scientific (English) language used in paper is poor. Many sentences need re-phrasing. Take help of native English speaker.

Reply:We had took help of native English speaker for polishing this article.

  • Line 348. How libraries prepared? Describe

Reply:We have detailed the process and method of library construction in revised manuscript

  • What parameters of HISAT2? Star is better aligner than HISAT2. Why HISAT2 used here?

Reply:The main parameter of HISAT2 in this paper is -- dta-p6 --max-intronlen 5. Through the following literature, HISAT2 had the highest splicing site validation rate in all samples. In terms of running speed, HISAT2 has a big advantage, about 2.5 times faster than STAR and about 100 times faster than TopHat. As a result, HISAT2 performs better in terms of accuracy and speed.

Sahraeian, S.M.E., Mohiyuddin, M., Sebra, R. et al. Gaining comprehensive biological insight into the transcriptome by performing a broad-spectrum RNA-seq analysis. Nat Commun 8, 59 (2017). https://doi.org/10.1038/s41467-017-00050-4

  • Section 4.3 re-write. Need to describe all the analyses.

Reply:We had corrected this part in revised manuscript.

  • Figure 9C. What is the significance of network?

Reply:The correlation network map can clarify the relationship between DEGs and DAMs, find key nodes, facilitate data mining, and simply understand the regulatory process from genes to metabolites. At the same time, the correlation intensity between genes and metabolites can be determined according to the network.

  • Discussion section needs to be re-written. It’s not going with flow of story

Reply:The discussion section has been sequentially modified according to the results section.

Round 2

Reviewer 1 Report

The vast majority of my concerns have been successfully addressed by the authors. So, I accept the manuscript for publication in its current form.

Reviewer 2 Report

Comments, 

The authors have taken steps to address some of the comments I raised, but issues with the English language remain. The manuscript still requires significant improvement and editing.

1)    Key finding, future directions are missing from conclusion.

2)    Line 277-279. Please re-check the sentence you wrote. (The sentence lacks coherence and meaning.)

3)    The entire paper still gives the impression of being a rough draft.

There are numerous instances where sentences require rephrasing. The language used in the paper continues to be a significant issue. I would strongly recommend rewriting the paper.

There are numerous instances where sentences require rephrasing. The language used in the paper continues to be a significant issue. I would strongly recommend rewriting the paper.

Author Response

1)    Key finding, future directions are missing from conclusion.

Reply: In this study, we performed transcriptome and metabolome sequencing of soybean leaves between nature light and shade conditions. Combined the results of multiple analyses, 12 candidate DEGs, such as ATP phosphoribosyl transferase (ATP-PRT2), phosphocho-line phosphatase (PEPC), AUXIN-RESPONSIVE PROTEIN (IAA17), PURPLE ACID PHOSPHATASE (PAP), were finally screened out and validated by qRT-PCR, such as ATP phosphoribosyl transferase (ATP-PRT2), phosphocholine phosphatase (PEPC), AUXIN-RESPONSIVE PROTEIN (IAA17), PURPLE ACID PHOSPHATASE (PAP), etc. In future, these candidate genes deserved more attention for the study on response mechanism of soybean shading.

2)    Line 277-279. Please re-check the sentence you wrote. (The sentence lacks coherence and meaning.)

Reply: Sorry for this mistake. We had corrected as “Transcriptome analysis of young leaves under two light environments in Cryptocarya concinna showed that the genes related to hormone had up-regulated expression under low light conditions (Zheng et al., 2021)”.

3)    The entire paper still gives the impression of being a rough draft.

Reply: On the premise of un-changing the overall idea and result of the article, We have checked and modified the whole text word by word. In revised manuscript, the words are more prepared and the sentences are more refined, rigorous and scientific.

There are numerous instances where sentences require rephrasing. The language used in the paper continues to be a significant issue. I would strongly recommend rewriting the paper.

Reply: Through many efforts, we have revised the language problems in revised manuscript.